# Tongue thickness measured by ultrasonography is associated with tongue pressure in the Japanese elderly

**Masahiro Nakamori**[1,2]*, **Eiji Imamura**[1], **Masako Fukuta**[3], **Keisuke Tachiyama**[1,2],
**Teppei Kamimura**[1,2], **Yuki Hayashi**[1,2], **Hayato Matsushima**[1], **Kanami Ogawa**[3],
**Masami Nishino**[3], **Akiko Hirata**[3], **Tatsuya Mizoue**[4], **Shinichi Wakabayashi**[4]

1 Department of Neurology, Suiseikai Kajikawa Hospital, Hiroshima, Japan, 2 Department of Clinical
Neuroscience and Therapeutics, Hiroshima University Graduate School of Biomedical and Health Sciences,
Hiroshima, Japan, 3 Department of Clinical Laboratory, Suiseikai Kajikawa Hospital, Hiroshima, Japan,
4 Department of Neurosurgery, Suiseikai Kajikawa Hospital, Hiroshima, Japan

* mnakamori1@gmail.com

org/10.1371/journal.pone.0230224

Romagna, ITALY

**Data Availability Statement:** All relevant data are
within the manuscript.

**Funding:** The authors received no specific funding
for this work.

## Abstract

The term "oral frailty" reflects the fact that oral health is associated with physical frailty and
mortality. The gold standard methods for evaluating the swallowing function have several
problems, including the need for specialized equipment, the risk of radiation exposure and
aspiration, and general physicians not possessing the requisite training to perform the
examination. Hence, several simple and non-invasive techniques have been developed for
evaluating swallowing function, such as those for measuring tongue pressure and tongue
thickness. The aim of this study was to investigate the relationship between tongue thick-
ness ultrasonography and tongue pressure in the Japanese elderly. We evaluated 254
elderly patients, who underwent tongue ultrasonography and tongue pressure measure-
ment. To determine tongue thickness, we measured the vertical distance from the surface
of the mylohyoid muscle to the tongue dorsum using ultrasonography. The results of the
analyses revealed that tongue thickness was linearly associated with tongue pressure in
both sexes. In male participants, dyslipidemia, lower leg circumference, and tongue pres-
sure were independently and significantly associated with tongue thickness. In female par-
ticipants, body mass index and tongue pressure were independently and significantly
associated with tongue thickness. The optimal cutoff for tongue thickness to predict the ton-
gue pressure of < 20 kPa was 41.3 mm in males, and 39.3 mm in females. In the Japanese
elderly, tongue thickness using ultrasonography is associated with tongue pressure. Tongue
thickness and tongue pressure, which are sensitive markers for oral frailty, decrease with
age. We conclude that tongue ultrasonography provides a less invasive technique for deter-
mining tongue thickness and predicts oral frailty for elderly patients.

## Introduction

In an aging society, such as present-day Japan, frailty is a critical issue related to morbidity as
well as mortality. Frailty is a common geriatric syndrome that embodies an elevated risk of

**Competing interests:** The authors have declared that no competing interests exist.

catastrophic declines in health and function in the elderly. Recent reports indicate that oral health is associated with physical frailty and mortality, hence the term "oral frailty" [1–3]. It can lead to dysphagia, dehydration, malnutrition, asphyxia, and aspiration pneumonia, which is one of the most life-threatening concerns for the elderly [4, 5].

The tongue is one of the most important organs related to oral frailty; swallowing dysfunction can lead to aspiration pneumonia and is therefore a critical concern. Quantitative evaluation of oral frailty often requires specialized instruments. Among them, the videofluoroscopic examination (VF) is one of the most reliable and is seen as the gold standard for evaluating the swallowing function. However, VF poses several problems because it requires equipment and carries the risk of radiation exposure and aspiration. A fiberoptic endoscopic evaluation of swallowing is also a gold standard method, which is routinely performed by otolaryngologists or rehabilitation doctors. However, many general physicians do not acquire the skills required to perform the examination. Hence, several simple and non-invasive techniques have been developed for evaluating swallowing function, such as those for measuring tongue pressure and tongue thickness [6–8]. For measuring tongue pressure, the instruments are mainly divided into the sensor sheets type and the balloon type. The sensor sheets-type device is attached to the hard palate of patients and they actually swallow liquid or food [9]. For the balloon-type device, tongue pressure is measured by having a patient raise the tongue and push on the hard palate [6–8]. Studies have reported that lower tongue pressure is a sensitive indicator for detecting swallowing dysfunction in patients who have had strokes and those with certain neurological disorders [10–12]. Of note, tongue pressure decreases with age and is significantly lower in frail Japanese elderly persons [7].

Tongue thickness is often measured by ultrasonography, which can generate a stable numerical value [13, 14]. It is reported that lower tongue pressure and decreasing tongue thickness measured by ultrasonography are associated with the oral preparatory and transit time measured using VF in amyotrophic lateral sclerosis patients [11, 13]. Tongue ultrasonography is an objective and non-invasive evaluation technique that carries no risk of aspiration.

Several reports have examined measures of tongue pressure in the general population and the elderly, but none have explored how tongue pressure relates to tongue thickness measured by ultrasonography. The aim of this study was to investigate the relationship between tongue thickness and tongue pressure in the Japanese elderly.

## Methods

### Ethics statement

The study protocols were approved by the ethics committee of Suiseikai Kajikawa Hospital and performed according to the guidelines of the national government based on the Helsinki Declaration of 1964. Written informed consent was obtained from all patients. All data analyses were blinded so no identifying information was revealed. The individual in this manuscript in Figs 1 and 2 has given written informed consent (as outlined in PLOS consent form) to publish the case details.

### Subjects

Consecutive outpatients who visited Suiseikai Kajikawa Hospital between February 1st and July 31st of 2019 were enrolled in this prospective study. We included those patients who consented to participate and were aged 65 years and older with lifestyle-related diseases (such as hypertension, diabetes mellitus, dyslipidemia, and chronic kidney disease). We excluded patients who had a history of otorhinolaryngologic disease (brain, facial, neck, pharyngeal, or laryngeal tumors) and/or neurodegenerative disease. Patients with oral abnormalities, such as

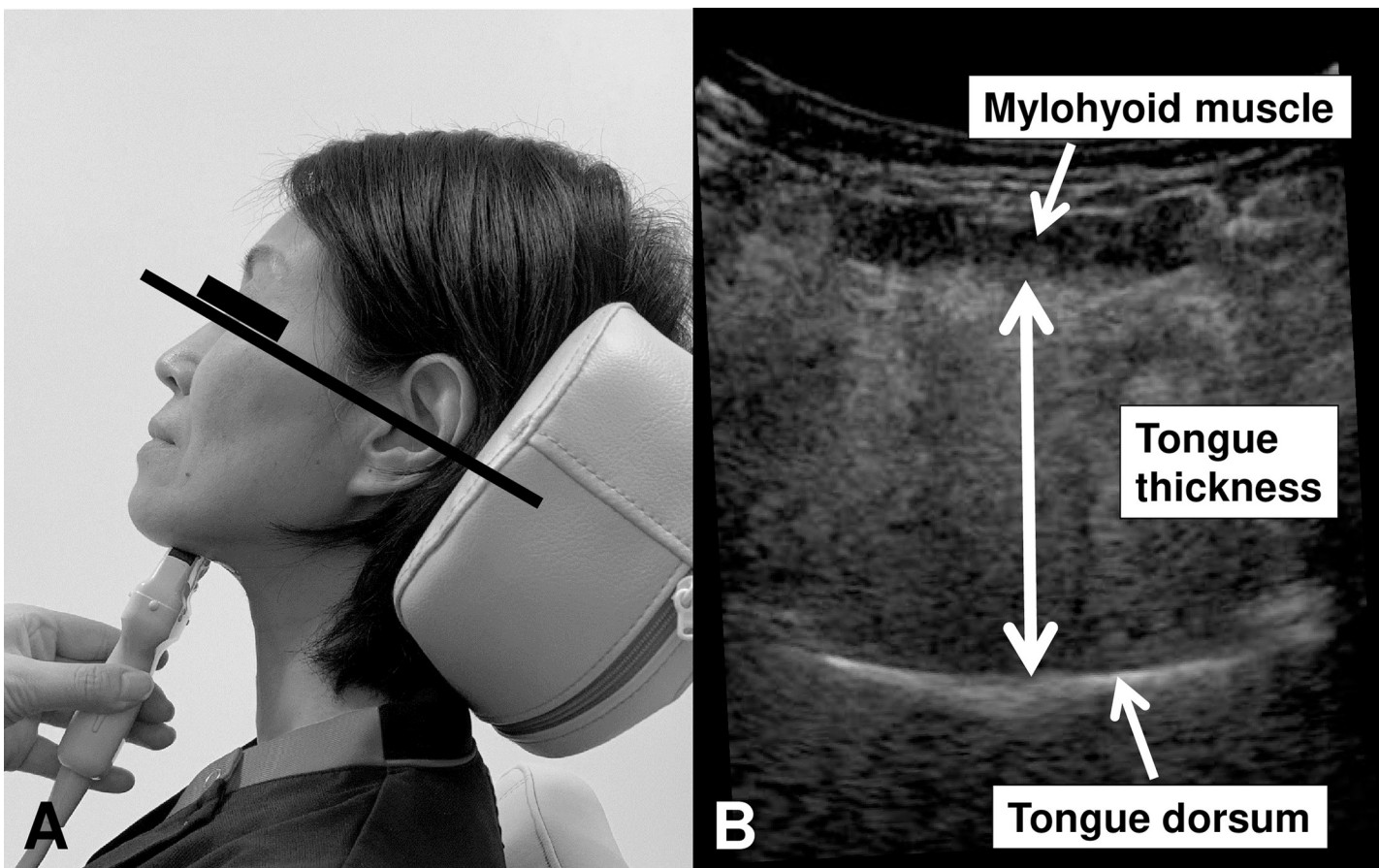

**Fig 1. Measurement of tongue thickness.** A) The subjects were examined in a 30° reclined position while seated. Tongue thickness was determined as the distance between the upper and lower surfaces of the lingual muscles in the center of the plane perpendicular to the Frankfurt horizontal plane in the frontal section. B) The vertical distance was measured from the surface of the mylohyoid muscle to the tongue dorsum.

tongue-tie; maxillofacial dysfunction, such as trismus; and those with extremity paralysis were also excluded.

### Tongue ultrasonography

The non-invasive ultrasound examinations were performed by a neurosonologist (M Nakamori) using a Noblus imaging system (Hitachi, Ltd., Tokyo, Japan). Tongue thickness was measured using a 2-8MHz convex array transducer according to a previously reported method, with the device placed under the chin of the participant. The subjects were examined in a 30° reclining position while seated. Tongue thickness was determined by measuring the distance between the upper and lower surfaces of the patient's lingual muscles in the center of the plane perpendicular to the Frankfurt horizontal plane of the frontal section (Fig 1A). This perpendicular plane intersects the distal surfaces of the bilateral mandibular second premolars. The vertical distance was measured from the surface of the mylohyoid muscle to the tongue dorsum (Fig 1B). This measurement was performed three times, and the mean value was defined as the tongue thickness for each participant. We confirmed the reliability of tongue ultrasonography by calculating the intra-rater and inter-rater reliability. For investigating intra-rater reliability, we measured the tongue thickness of the normal subject three times per day; their mean value was defined as the tongue thickness for the day. These measurements

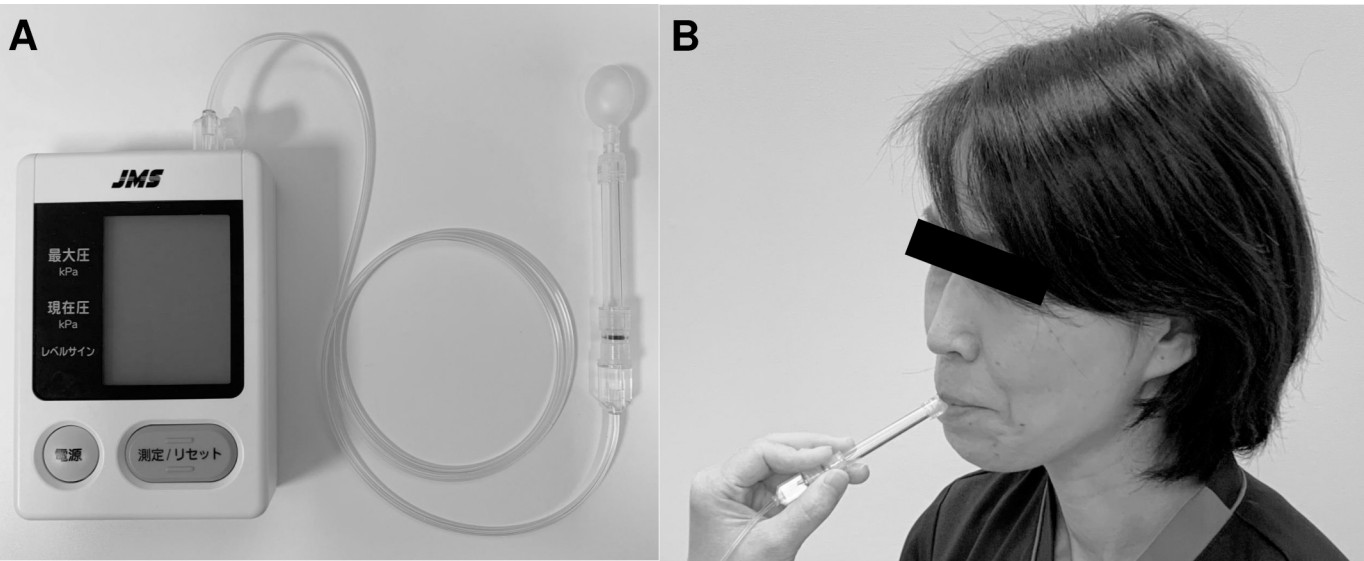

**Fig 2. Measurement of tongue pressure.** A) The balloon-type equipment consists of a disposable oral probe, an infusion tube as a connector, and a recording device. B) The subjects were asked to place the balloon in their mouths, holding the plastic pipe at the midpoint of their central incisors with closed lips. The subjects were asked to raise their tongue and compress the small balloon with their palate at maximum voluntary effort for 7 seconds.

were repeated for ten days, and the resulting coefficient of variation was 1.59%. To investigate the inter-rater variability, two investigators (M Nakamori and MF) independently measured the tongue thickness of the same 23 normal subjects, and the resulting coefficient of variation was under 1.74%. Bland-Altman analysis was performed, and systematic (fixed and proportional) errors were not detected.

## Tongue pressure measurement

Clinical technicians (KO, M Nishino, and AH) measured tongue pressure independently using balloon-type equipment (TPM-01; JMS Co. Ltd., Hiroshima, Japan) on the same day when the patients underwent measurement of tongue thickness using ultrasonography. The balloon-type equipment consisted of a disposable oral probe, an infusion tube as a connector, and a recording device (Fig 2A). For tongue pressure measurement, the subjects were placed in a relaxed sitting position and asked to place the balloon in their mouths, holding the plastic pipe at the midpoint of their central incisors with closed lips. The subjects were asked to maintain this position as clinicians adjusted the probe and confirmed that it was in the correct position. The subjects were then asked to raise their tongue and compress the small balloon with their palate at maximum voluntary effort for seven seconds as described previously (Fig 2B) [6, 15]. This measurement was performed three times with the subjects resting for approximately 30 seconds and rinsing their mouths between each measurement. The highest value from the three measurements was defined as the tongue pressure for each subject. The reliability of intraindividual measurement has been previously reported [10, 16].

## Data acquisition

Patient characteristics, including age, gender, body mass index (BMI), past history of comorbidities (hypertension, diabetes mellitus, dyslipidemia, chronic kidney disease), grip power, lower leg circumference, serum albumin, tongue pressure, and tongue thickness were evaluated. Hypertension was defined as the use of anti-hypertensive medication or confirmed blood

pressure of $\geq$ 140/90 mmHg at rest. Diabetes mellitus was defined as a glycated hemoglobin level of $\geq$ 6.5%, fasting blood glucose level of $\geq$ 126 mg/dl, or use of anti-diabetes medication. Dyslipidemia was defined as a total cholesterol level of $\geq$ 220 mg/dl, low-density lipoprotein cholesterol level of $\geq$ 140 mg/dl, high-density lipoprotein cholesterol level of < 40 mg/dl, tri-glyceride levels of $\geq$ 150 mg/dl, or use of anti-hyperlipidemia medication. Renal functioning was calculated with the estimated glomerular filtration rate (eGFR) using a revised equation for the Japanese population as follows: eGFR (ml min−1 1.73 m−2) = 194 × (serum creatinine) −1.094 × (age)−0.287 × 0.739 (for women) [17]. Chronic kidney disease was defined as an eGFR < 60 ml min−1 1.73 m−2. Grip power was measured for both sides and the mean value was used for analysis. Lower leg circumference was measured at the thickest place at both sides and the mean value was used for analysis. Additionally, swallowing was evaluated using the Food Intake LEVEL Scale (FILS), which was administered by two physicians (YH and HM) [18]. FILS is a 10-point observer-rating scale to measure the severity of swallowing dysfunction. Its convergent validity and intra-rater and inter-rater reliability have been established with the Functional Oral Intake Scale.

## Statistical analysis

The data were expressed as the mean ± standard deviation for continuous variables and frequencies and percentages for discrete variables. Statistical analysis was performed using JMP 13 statistical software (SAS Institute Inc., Cary, NC, USA). The statistical significance of inter-group differences was assessed using unpaired $t$-tests or $\chi^2$ tests as appropriate. We calculated the required sample size according to past investigations for the tongue thickness and tongue pressure in amyotrophic lateral sclerosis [11, 13]. Based on an alpha level = 0.05, and power = 0.80, we estimated that we would require a total of n = 212 participants. The baseline data for the subjects were analyzed, and two-step strategies were employed to assess the relative importance of variables in their association with tongue thickness using least square linear regression analysis. First, a univariate analysis was performed. Then, a multifactorial least-square linear regression analysis was performed with selected factors that had $p < 0.20$ on the univariate analysis. Tongue thickness and tongue pressure were compared by five-year age increments. The data were analyzed with a one-way analysis of variance and Tukey's honestly significant difference (HSD) test. Receiver operating characteristic (ROC) analysis was performed to determine the tongue thickness predicting a tongue pressure < 20kPa, which suggests swallowing dysfunction. We considered $p < 0.05$ as statistically significant.

## Results

We evaluated 254 elderly patients, whose backgrounds are shown in Table 1. Tongue thickness, tongue pressure, grip power, and lower leg circumference were all markedly different between the male group and the female group. To account for a disproportionate physique owing to normal differences between males and females, results were analyzed separately by sex.

Scatter plots were used to display tongue thickness and tongue pressure by sex. Linear regression analyses indicated that tongue thickness was linearly associated with tongue pressure in both sexes (male; coefficient 0.202, 95% confidence interval 0.182–0.223, $p < 0.001$; Fig 3A, female; coefficient 0.202, 95% confidence interval 0.182–0.223, $p < 0.001$; Fig 3B).

The potential factors associated with tongue thickness and the FILS scores are listed in Table 1 were evaluated using multifactorial regression analysis by sex. In the male group, dyslipidemia, lower leg circumference, and tongue pressure were independently significant in their association with tongue thickness (adjusted $R^2$ = 0.653, $p < 0.001$, n = 163) (Table 2A). In the

**Table 1. Japanese elder participants' health-related factors in tongue thickness study.**

| | All | Male | Female | |
|---|---|---|---|---|
| | n = 254 | n = 163 | n = 91 | p-value |
| Age | 77.9 ± 6.3 | 76.9 ± 5.8 | 79.5 ± 6.7 | 0.001* |
| Body mass index, kg/m$^2$ | 23.2 ± 3.0 | 23.3 ± 2.8 | 23.2 ± 3.2 | 0.776 |
| Hypertension, n (%) | 207 (81.5) | 129 (79.1) | 78 (85.7) | 0.196 |
| Diabetes mellitus, n (%) | 51 (20.0) | 35 (21.5) | 16 (17.6) | 0.458 |
| Dyslipidemia, n (%) | 172 (67.7) | 104 (63.8) | 68 (74.7) | 0.074 |
| Chronic kidney disease, n (%) | 48 (18.9) | 33 (20.3) | 15 (16.5) | 0.463 |
| Grip power, kg | 24.2 ± 8.7 | 28.8 ± 6.9 | 15.9 ± 4.5 | <0.001* |
| Lower leg circumference, cm | 33.6 ± 3.3 | 34.4 ± 3.2 | 32.1 ± 2.8 | <0.001* |
| Serum albumin, g/dl | 4.1 ± 0.3 | 4.1 ± 0.3 | 4.1 ± 0.4 | 0.345 |
| Tongue pressure, kpa | 35.7 ± 10.6 | 37.4 ± 10.2 | 32.5 ± 10.5 | <0.001* |
| Tongue thickness, mm | 41.9 ± 2.8 | 42.4 ± 2.6 | 40.9 ± 2.8 | <0.001* |
| Food Intake LEVEL Scale | 10 (8, 10) | 10 (8, 10) | 10 (8, 10) | 0.510 |

$^*$p < 0.05.

female group, body mass index and tongue pressure were independently significant in their association with tongue thickness (adjusted R$^2$ = 0.707, $p$ < 0.001, n = 91) (Table 2B).

Tongue thickness was compared by five-year age increments for each sex (Fig 4). These data suggested that tongue thickness significantly decreased by age in each sex ($p$ < 0.001). In addition, tongue pressure was compared by five-year age increments for each sex (Fig 5). These data suggested that tongue pressure also significantly decreased by age in each sex ($p$ < 0.001). Tukey's HSD tests revealed that both tongue thickness and tongue pressure remarkably decreased in patients over 85 years old.

Several previous reports suggested that using the JMS balloon-type device, the tongue pressure of patients with swallowing dysfunction was observed to be approximately < 20 kPa [10–

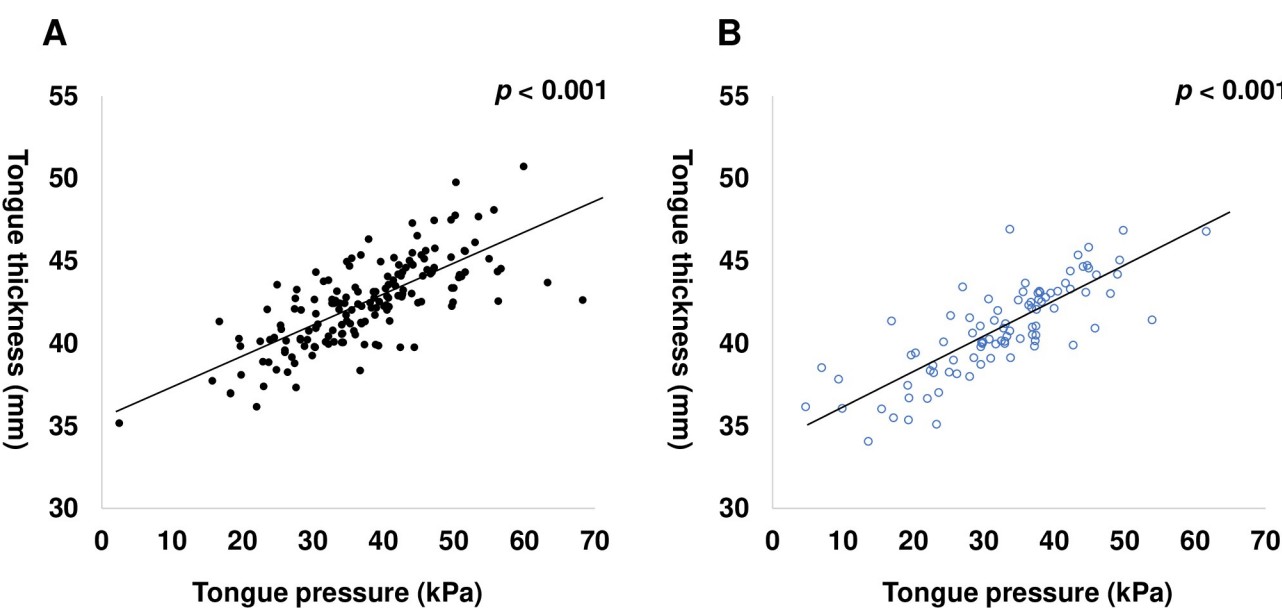

**Fig 3. The association of tongue thickness with tongue pressure.** In the elderly, tongue thickness is shown to be associated with tongue pressure A) in males ($p$ < 0.001) and B) in females ($p$ < 0.001); •, male; ○, female.

**Table 2. Factors influencing tongue thickness in Japanese elders.**

| A | | | | |
|---|---|---|---|---|
| Male | Univariate | Multivariate | | |
| | p-value | coefficient | 95% CI | p-value |
| Age | <0.001 | -0.001 | -0.050–0.047 | 0.962 |
| Body mass index | <0.001 | 0.070 | -0.057–0.198 | 0.277 |
| Hypertension | 0.937 | | | |
| Diabetes mellitus | 0.174 | -0.266 | -0.564–0.032 | 0.080 |
| Dyslipidemia | 0.032 | -0.264 | -0.519 - -0.001 | 0.043* |
| Chronic kidney disease | 0.406 | | | |
| Grip power | <0.001 | 0.032 | -0.011–0.075 | 0.141 |
| Lower leg circumference | <0.001 | 0.192 | 0.065–0.319 | 0.003* |
| Serum albumin | <0.001 | 0.541 | -0.391–1.473 | 0.253 |
| Tongue pressure | <0.001 | 0.142 | 0.114–0.169 | <0.001* |
| B | | | | |
| Female | Univariate | Multivariate | | |
| | p-value | coefficient | 95% CI | p-value |
| Age | <0.001 | -0.004 | -0.067–0.059 | 0.896 |
| Body mass index | <0.001 | 0.195 | 0.053–0.337 | 0.008* |
| Hypertension | 0.776 | | | |
| Diabetes mellitus | 0.336 | | | |
| Dyslipidemia | 0.989 | | | |
| Chronic kidney disease | 0.474 | | | |
| grip power | <0.001 | 0.073 | -0.019–0.164 | 0.118 |
| lower leg circumference | <0.001 | 0.082 | -0.094–0.258 | 0.359 |
| Serum albumin | 0.077 | 0.394 | -0.569–1.357 | 0.419 |
| Tongue pressure | <0.001 | 0.171 | 0.135–0.207 | <0.001* |

*p < 0.05 on multivariate analysis.

12]. The optimal cutoff for tongue thickness to predict the tongue pressure < 20 kPa was 41.3 mm in male from the ROC analysis ($\chi^2$ = 24.48, $p$ < 0.001, sensitivity 100.0%, specificity 68.4%, AUC = 0.91), and 39.3 mm in female from the ROC analysis ($\chi^2$ = 32.29, $p$ < 0.001, sensitivity 91.7%, specificity 82.3%, AUC = 0.92).

Finally, the potential factors associated with the FILS scores were evaluated using multifactorial regression analysis by sex. In both the male and female group, only tongue pressure was significantly associated with the FILS scores (male; adjusted $R^2$ = 0.592, $p$ < 0.001, n = 163: Table 3A, female; adjusted $R^2$ = 0.764, $p$ < 0.001, n = 9: Table 3B).

## Discussion

In the present study, we investigated the tongue thickness of elder Japanese patients using ultrasonography. We focused on reasonably healthy elderly patients who have a few lifestyle diseases. Our results showed that tongue thickness was independently associated with tongue pressure and lower leg circumference in males, and tongue pressure and body mass index in females. A decrease of lower leg circumference or body mass index is also known to be related to sarcopenia which leads to frailty. In addition, it has been reported that tongue pressure is significantly reduced in frail elderly Japanese persons [7]. Hence, decreased tongue thickness may also be one of the signs of frailty.

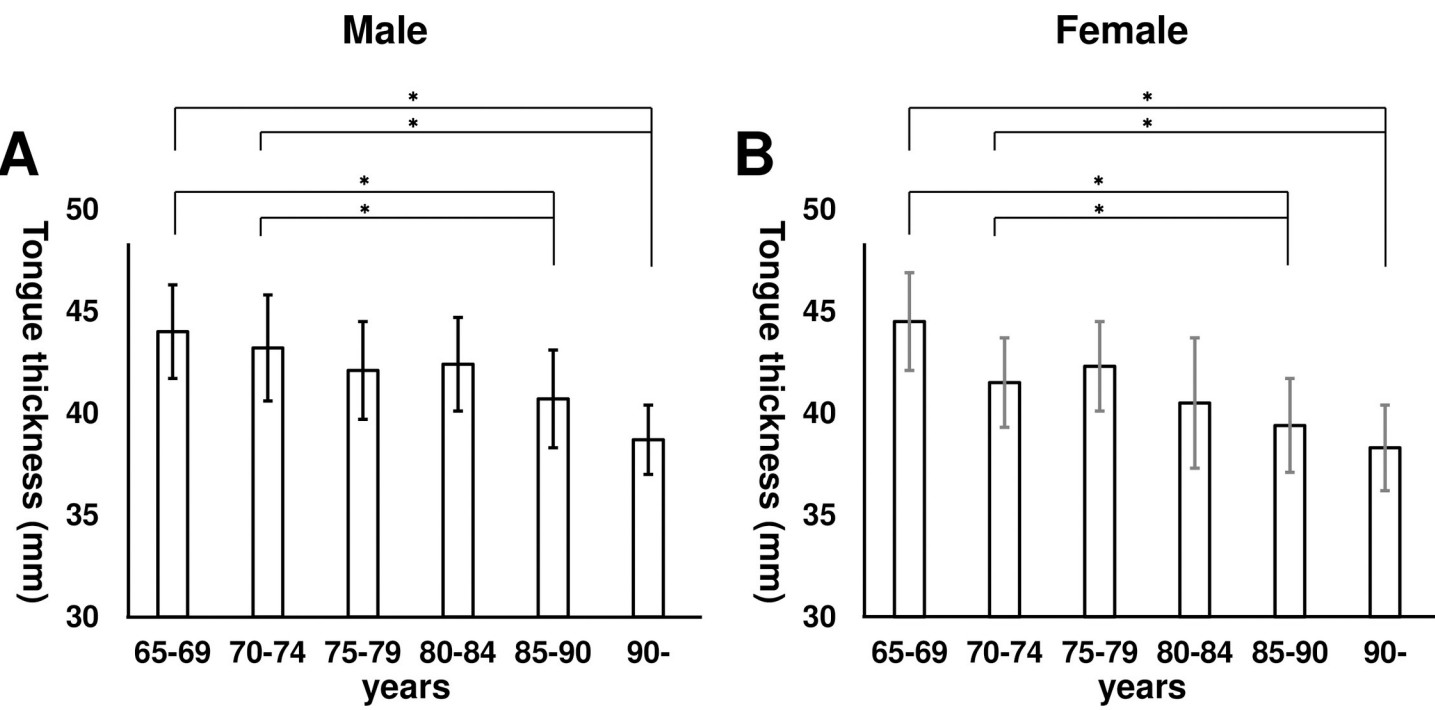

**Fig 4. Tongue thickness by age and sex.** A) Tongue thickness by five-year age increments in males. Tongue thickness significantly decreased with increased age ($p < 0.001$). B) Tongue thickness by five-year age increments in females. Tongue thickness was significantly decreased with increased age ($p < 0.001$). *$p < 0.05$ by Tukey's honestly significant difference test.

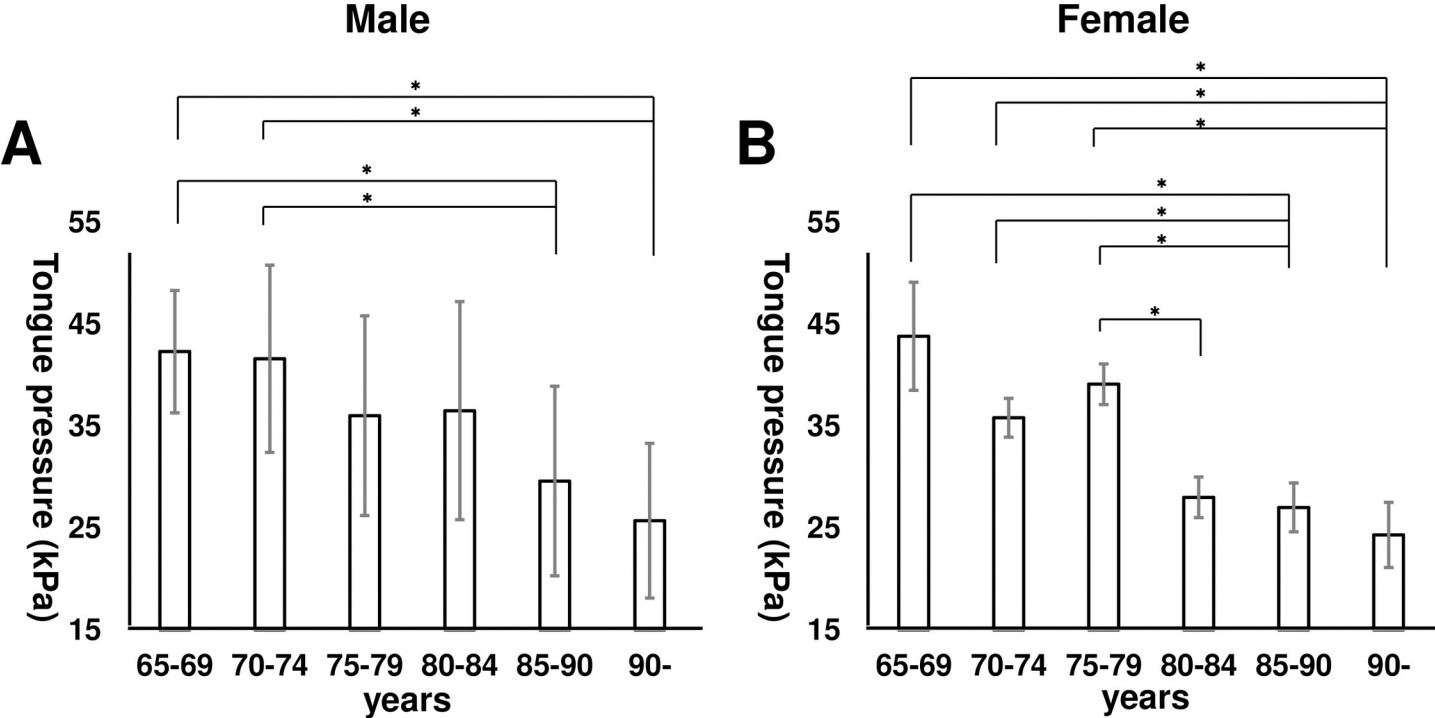

**Fig 5. Tongue pressure by age and sex.** A) Tongue pressure by five-year age increments in males. Tongue pressure significantly decreased with increased age ($p < 0.001$). B) Tongue pressure by five-year age increments in females. Tongue pressure significantly decreased with increased age ($p < 0.001$). *$p < 0.05$ by Tukey's honestly significant difference test.

**Table 3. Factors associated with the scores for the food intake LEVEL scale.**

| A | | | | |
|---|---|---|---|---|
| Male | Univariate | Multivariate | | |
| | p-value | coefficient | 95% CI | p-value |
| Age | 0.005 | -0.001 | -0.013–0.012 | 0.909 |
| Body mass index | 0.059 | 0.028 | -0.062–0.007 | 0.116 |
| Hypertension | 0.209 | | | |
| Diabetes mellitus | 0.359 | | | |
| Dyslipidemia | 0.482 | | | |
| Chronic kidney disease | 0.876 | | | |
| Grip power | 0.002 | 0.002 | -0.010–0.013 | 0.757 |
| Lower leg circumference | <0.001 | 0.026 | -0.009–0.061 | 0.148 |
| Serum albumin | 0.010 | 0.032 | -0.218–0.282 | 0.800 |
| Tongue pressure | <0.001 | 0.013 | 0.003–0.022 | 0.009* |
| Tongue thickness | <0.001 | 0.028 | -0.014–0.070 | 0.191 |
| B | | | | |
| Female | Univariate | Multivariate | | |
| | p-value | coefficient | 95% CI | p-value |
| Age | 0.001 | -0.005 | -0.021–0.010 | 0.514 |
| Body mass index | 0.269 | | | |
| Hypertension | 0.100 | 0.050 | -0.068–0.168 | 0.400 |
| Diabetes mellitus | 0.690 | | | |
| Dyslipidemia | 0.748 | | | |
| Chronic kidney disease | 0.225 | | | |
| Grip power | 0.003 | 0.008 | -0.014–0.029 | 0.467 |
| Lower leg circumference | 0.085 | 0.013 | -0.048–0.022 | 0.465 |
| Serum albumin | 0.220 | | | |
| Tongue pressure | <0.001 | 0.030 | 0.017–0.043 | <0.001* |
| Tongue thickness | <0.001 | 0.020 | -0.073–0.032 | 0.438 |

*$p < 0.05$ on multivariate analysis.

Tongue thickness and tongue pressure significantly decreased with age in each sex. In our analysis, the decrease was most remarkable in participants over 85 years old. These results allude to the rate of progression of tongue muscle atrophy due to aging. We generally judge tongue atrophy by visual examination, but it is fairly subjective, depending on the opinion of the examiner. In contrast, measurement with ultrasonography is quantitative and objective. Several previous reports have suggested that the tongue pressure of patients with swallowing dysfunction was observed to be approximately < 20 kPa [10–12] using the JMS balloon-type device. ROC analysis of the results of our study revealed that the optimal cutoff for tongue thickness to predict tongue pressure of < 20 kPa is 41.3 mm for males and 39.3 mm for females. There are three main types of balloon-based tongue pressure measurement devices: the KayPENTAX device, the Iowa Oral Performance Instrument (IOPI), and the JMS Co. device. The JMS device, which is commonly used in Japan, shows lower values than the other balloon-based devices, but said values can be correlated linearly [6]. The IOPI is often used internationally, and shows the relationship between tongue pressure and age, which is consistent with our findings [19].

We compared the actual swallowing state and baseline factors using FILS. FILS was associated with tongue pressure but not tongue thickness. In this study, all patients scored 8 or more

on the FILS, which indicates the patients can eat three meals by excluding food that is particularly difficult to swallow. Tongue pressure is superior to tongue thickness to detect early or mild swallowing dysfunction.

One serious concern related to oral frailty is that elderly people often develop aspiration pneumonia. Aspiration pneumonia is caused by many factors such as oral environment, swallowing dysfunction, decreased cough reflex, and immunodepression. These factors are associated with oral frailty [20]. Additionally, decreased tongue strength is associated with sarcopenia and contributes to oral frailty [21]. A reduction in tongue strength has been associated with an increased risk for aspiration as it increases the likelihood of bolus retention in the pharynx [22]. While tongue pressure as measured at approximately 20 kPa is a cutoff value for swallowing dysfunction, many people who meet that description do not report significant oral dysfunction or experience dysphagia [11, 12]. For early detection, measurements of tongue thickness and tongue pressure should be performed as routine screening measures in the general elderly population. However, the measurement of tongue pressure requires a patient to be capable of comprehending the process and to actively participate while following instructions. Patients with dementia, for example, may not be able to undergo tongue pressure measurements. For these patients, tongue thickness measurements might be useful as a more suitable screening test. Rehabilitation for raising tongue pressure has been developed and reported as effective for the general elderly population [23]. Moreover, it has been shown that the maintenance of body weight by nutritional intervention may inhibit the progression of tongue atrophy [13]. Nutritional care is also important for preventing oral and systemic frailty.

This study, while encouraging, also has limitations. First, the measurements were not compared with actual oral dysfunction or gold standard methods such as VF. In amyotrophic lateral sclerosis patients, it has been reported that tongue thickness and tongue pressure show an association in VF temporary analysis, especially in oral preparatory and transit time [11, 13]. However, in the general elderly, there are only limited opportunities to perform VF. In this study, we evaluated the FILS and investigated its relationship with tongue pressure and tongue thickness. Second, in the present study, tongue thickness of older participants was not compared with that of younger healthy subjects. Data exists regarding tongue pressure, as it has been measured in all ages. Further studies are needed to measure the tongue thickness in a general population including all ages. Third, we could not evaluate sleep apnea in the subjects of this study. It has been reported that people with sleep apnea have decreased tongue strength and increased tongue thickness due to excessive fat deposition, which causes tongue collapse [24, 25]. In the present study, there were 3 subjects whose BMI was over 30 kg/m$^2$ and were not diagnosed with obstructive sleep apnea hypopnea syndrome (OSAHS). However, we did not perform a polysomnography on all subjects. In future research, sleep apnea should be considered when studying the swallowing function.

## Conclusion

In the Japanese elderly, tongue thickness, as measured using ultrasonography, is associated with tongue pressure. Decreases in both tongue thickness and tongue pressure occur in aging patients and may be sensitive markers of oral frailty. Oral frailty decreases tongue pressure and tongue thickness, which leads to systemic frailty and, finally, to increased morbidity and mortality. Early detection using such instruments might be important in preventing the progression of frailty.

## Acknowledgments

We would like to sincerely thank the staff at the Suiseikai Kajikawa Hospital for their technical assistance.

## Author Contributions

**Conceptualization:** Masahiro Nakamori, Eiji Imamura, Masako Fukuta, Keisuke Tachiyama, Yuki Hayashi, Masami Nishino, Akiko Hirata, Shinichi Wakabayashi.

**Data curation:** Masahiro Nakamori, Masako Fukuta, Keisuke Tachiyama, Teppei Kamimura, Yuki Hayashi, Hayato Matsushima, Kanami Ogawa, Akiko Hirata.

**Formal analysis:** Masahiro Nakamori, Akiko Hirata.

**Investigation:** Masahiro Nakamori, Masako Fukuta, Kanami Ogawa, Masami Nishino, Akiko Hirata.

**Methodology:** Masahiro Nakamori, Masako Fukuta, Teppei Kamimura, Hayato Matsushima, Kanami Ogawa, Masami Nishino, Akiko Hirata.

**Project administration:** Masahiro Nakamori.

**Supervision:** Eiji Imamura, Akiko Hirata, Tatsuya Mizoue, Shinichi Wakabayashi.

**Validation:** Eiji Imamura, Tatsuya Mizoue, Shinichi Wakabayashi.

**Writing – original draft:** Masahiro Nakamori, Masako Fukuta, Keisuke Tachiyama, Teppei Kamimura, Yuki Hayashi, Masami Nishino, Shinichi Wakabayashi.

**Writing – review & editing:** Masahiro Nakamori, Eiji Imamura, Tatsuya Mizoue, Shinichi Wakabayashi.

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
