## [Decision Letter · Decision Letter 0]

11 Jun 2020

PONE-D-20-05211

Tongue thickness measured by ultrasonography is associated with tongue pressure in the Japanese elderly

PLOS ONE

Dear Dr. Nakamori,

Thank you for submitting your manuscript to PLOS ONE. After careful consideration, we feel that it has merit but does not fully meet PLOS ONE’s publication criteria as it currently stands. Therefore, we invite you to submit a revised version of the manuscript that addresses the points raised during the review process.

We look forward to receiving your revised manuscript.

Kind regards,

Giovanni Cammaroto

Academic Editor

PLOS ONE

Journal Requirements:

2. We note that Figures 1 and 2 include an image of a [patient / participant / in the study]. 

Additional Editor Comments (if provided):

Reviewers' comments:

Reviewer's Responses to Questions

**Comments to the Author**

1. Is the manuscript technically sound, and do the data support the conclusions?

Reviewer #1: Yes

Reviewer #2: Yes

2. Has the statistical analysis been performed appropriately and rigorously? 

Reviewer #1: Yes

Reviewer #2: Yes

3. Have the authors made all data underlying the findings in their manuscript fully available?

Reviewer #1: No

Reviewer #2: Yes

4. Is the manuscript presented in an intelligible fashion and written in standard English?

Reviewer #1: Yes

Reviewer #2: Yes

5. Review Comments to the Author

Reviewer #1: The authors investigated the relationship between tongue thickness and tongue pressure in 254 Japanese elderly persons. The structure is adequate, the methods/results are well-written. Some points have to be addressed.

1. Reduce the size of the introduction. Focus on the importance of the topic and why it is important to conduct this study.

2. Some points of the discussion (sentences about aspirations, etc.) need to be replace in the introduction to shed light the importance of the topic. Please, in the discussion only focus of the discussion of your results.

3. The main weakness of this study is the lack of swallowing and aspiration evaluations. It's interesting to study the tongue size & pressure but it's needed for a practical idea. Here, there were no relationship with dysphagia or aspiration, which seem to be very important point associated with elderly frailty and morbi-mortality in case of disorder.

Reviewer #2: This study is very interesting. The authors evaluate the thickness of the tongue as an expression of “oral frailty” and as possible swallowing disorder. I think that the role of tongue in dysphagia should be better described in the text. In particular describe how a reduced tongue strength can cause inhalation risk.

The statistical analysis been performed appropriately and rigorously. It would be interesting to know if the sample size estimation and power analysis was calculated. The limitations of the study are well described.

6. PLOS authors have the option to publish the peer review history of their article (what does this mean?). If published, this will include your full peer review and any attached files.

Reviewer #1: No

Reviewer #2: No

---

## [Author Response · Author response to Decision Letter 0]

26 Jun 2020

We appreciate your advice. The manuscript has been revised as follows.

Response to the editor

Comment 1: Please ensure that your manuscript meets PLOS ONE's style requirements, including those for file naming. The PLOS ONE style templates can be found at https://journals.plos.org/plosone/s/file?id=wjVg/PLOSOne_formatting_sample_main_body.pdf and https://journals.plos.org/plosone/s/file?id=ba62/PLOSOne_formatting_sample_title_authors_affiliations.pdf

Response 1: We have ensured that the manuscript meets PLOS ONE's style requirements.

Comment 2: We note that Figures 1 and 2 include an image of a [patient / participant / in the study]. As per the PLOS ONE policy (http://journals.plos.org/plosone/s/submission-guidelines#loc-human-subjects-research) on papers that include identifying, or potentially identifying, information, the individual(s) or parent(s)/guardian(s) must be informed of the terms of the PLOS open-access (CC-BY) license and provide specific permission for publication of these details under the terms of this license. Please download the Consent Form for Publication in a PLOS Journal (http://journals.plos.org/plosone/s/file?id=8ce6/plos-consent-form-english.pdf). The signed consent form should not be submitted with the manuscript, but should be securely filed in the individual's case notes. Please amend the methods section and ethics statement of the manuscript to explicitly state that the patient/participant has provided consent for publication: “The individual in this manuscript has given written informed consent (as outlined in PLOS consent form) to publish these case details”.　If you are unable to obtain consent from the subject of the photograph, you will need to remove the figure and any other textual identifying information or case descriptions for this individual.

Response 2: We downloaded the Consent Form for Publication and the signed consent form is filed securely with our research documents. We amended the “Ethics Statement” in the “Methods” section to include the statement you provided.

Page 6, Lines 82-84

The individual in this manuscript in Figures 1 and 2 has given written informed consent (as outlined in PLOS consent form) to publish the case details. 

Comment 3: Please include captions for your Supporting Information files at the end of your manuscript, and update any in-text citations to match accordingly. Please see our Supporting Information guidelines for more information: http://journals.plos.org/plosone/s/supporting-information.

Response 3: We included captions for the supporting information files at the end of the manuscript but before the references, due to our use of reference management software and the desire to have our captions recognized as text and not citations. No in-text citations were provided for the supporting information.

Response to the reviewers

Reviewer #1: The authors investigated the relationship between tongue thickness and tongue pressure in 254 Japanese elderly persons. The structure is adequate, the methods/results are well-written. Some points have to be addressed.

Comment 1: Reduce the size of the introduction. Focus on the importance of the topic and why it is important to conduct this study.

Response 1: We focused the introduction on the importance of the topic and deleted the sentences containing general information. Because another reviewer recommended adding an explanation of swallowing evaluation and tongue pressure to the introduction, we added the explanation.

Comment 2: Some points of the discussion (sentences about aspirations, etc.) need to be replace in the introduction to shed light the importance of the topic. Please, in the discussion only focus of the discussion of your results.

Response 2: We revised the introduction to shed light on the importance of the topic. We also rewrote the discussion section to focus solely on the discussion of our results. 

Comment 3: The main weakness of this study is the lack of swallowing and aspiration evaluations. It's interesting to study the tongue size & pressure but it's needed for a practical idea. Here, there were no relationship with dysphagia or aspiration, which seem to be very important point associated with elderly frailty and morbi-mortality in case of disorder.

Response 3: As we mentioned in the limitations, we could not perform gold standard methods such as VF because the subjects were outwardly almost healthy patients. However, we evaluated participants using the ‘Food Intake LEVEL Scale (FILS),’ which is a scale for oral intake. We added the FILS as baseline data and re-analyzed the results. We show the relationship between FILS and baseline data in Table 3, and explained the relationship in the “Results” section. Below is the relevant text from the “Methods” and “Results” sections:

Page 10, Lines 159-162

Additionally, swallowing was evaluated using the Food Intake LEVEL Scale (FILS), which was administered by two physicians (YH and HM) [18]. FILS is a 10-point observer-rating scale to measure the severity of swallowing dysfunction, with lower scores indicating higher levels of dysfunction. Its convergent validity and intra-rater and inter-rater reliability have been established with the Functional Oral Intake Scale.

Page 15, Lines 237-240

Finally, the potential factors associated with the FILS scores were evaluated using multifactorial regression analysis by sex. In both the male and female group, only tongue pressure was significantly associated with the FILS scores (male; adjusted R2 = 0.592, p < 0.001, n = 163: Table 3A, female; adjusted R2 = 0.764, p < 0.001, n = 9: Table 3B). 

Reviewer #2: This study is very interesting. The authors evaluate the thickness of the tongue as an expression of “oral frailty” and as possible swallowing disorder. I think that the role of tongue in dysphagia should be better described in the text. In particular describe how a reduced tongue strength can cause inhalation risk.

The statistical analysis been performed appropriately and rigorously. It would be interesting to know if the sample size estimation and power analysis was calculated. The limitations of the study are well described.

In this study, the authors evaluate the thickness of the tongue as an expression of “oral frailty” and as possible swallowing disorder. 

Abstract.

Comment 1: Page 3, lines 33, 34. The sentence: “The tongue ultrasonography provides a less invasive technique for determining tongue thickness and related aspiration risks for elderly patients”. The authors should better explain or change the second part of this claim. Does the tongue ultrasonography provide a technique for determining aspiration risks?

Response 1: We appreciate your suggestion. We rewrote the sentence as follows.

Page 3, lines 38-40 

We conclude that tongue ultrasonography provides a less invasive technique for determining tongue thickness and predicts oral frailty for elderly patients.

Introduction.

Comment 2: Page 4, lines 56, 57. VF is not performed in some centers, but currently it is performed routinely in many hospitals. You should change the sentence: “…However, VF is a specialized method that is not routinely performed with the general population...”. 

Response 2: We appreciate your suggestion. We rewrote the sentence as follows.

Page 4, Lines 53-54

However, VF poses several problems because it requires equipment and carries the risk of radiation exposure and aspiration.

Comment 3: Page 4, lines 57, 58. “Hence, several simple and non-invasive techniques have been developed for evaluating swallowing function…”. A non invasive techniques for evaluating swallowing function is the Fiberoptic Endoscopic Evaluation of Swallowing (FEES) that is a gold standard for evaluation of pharyngeal phase of swallowing. You should add this exam with its bibliography to the introduction.

Response 3: We appreciate your suggestion. We rewrote the sentence as follows.

Pages 4-5, Lines 54-59

A fiberoptic endoscopic evaluation of swallowing is also a gold standard method, which is routinely performed by otolaryngologists or rehabilitation doctors. However, many general physicians do not acquire the skills required to perform the examination. Hence, several simple and non-invasive techniques have been developed for evaluating swallowing function, such as those for measuring tongue pressure and tongue thickness.

Comment 4: Page 4, lines 59, 60. You should describe some instrumental methods of measuring the tongue pressure with the relative bibliography.

Response 4: We appreciate your suggestion. We added the description as follows.

Pages 4-5, Lines 59-63

For measuring tongue pressure, the instruments are mainly divided into the sensor sheets type and the balloon type. The sensor sheets-type device is attached to the hard palate of patients and they actually swallow liquid or food [9]. For the balloon-type device, tongue pressure is measured by having a patient raise the tongue and push on the hard palate [6-8].

Mathods. Subjects.

Comment 5: Page 5, lines 82-85. What are the inclusion and exclusion criteria? What does it mean: “chronic disease”? “We excluded patients who had a history of otorhinolaryngologic disease”. Which ENT diseases have been excluded?

Response 5: Chronic diseases are lifestyle-related diseases (such as hypertension, diabetes mellitus, dyslipidemia). Otorhinolaryngologic disease is a pharyngeal or laryngeal tumor. 

We rewrote the inclusionary and exclusionary criteria:

Page 6, Lines 87-91

We included those patients who consented to participate and were aged 65 years and older with lifestyle-related diseases (such as hypertension, diabetes mellitus, dyslipidemia). We excluded patients who had a history of otorhinolaryngologic disease (pharyngeal or laryngeal tumor) and/or neurodegenerative disease. Patients with paralysis were also excluded. 

Statistical analysis.

Comment 6: In your study how the sample size estimation and power analysis was calculated?

Response 6: We appreciate your suggestion about statistical methods to make solid conclusions. We calculated the sample size according to the past investigations for the tongue thickness and tongue pressure in amyotrophic lateral sclerosis. We rewrote sentences in the “Statistical analysis” subsection of the “Methods” section as follows.

Pages 10-11, Lines 167-170

We calculated the required sample size according to past investigations for the tongue thickness and tongue pressure in amyotrophic lateral sclerosis [11,13]. Based on an alpha level = 0.05, and power = 0.80, we estimated that we would require a total of n = 212 participants.

Discussion.

Comment 7: Page 15, lines 250, 251. “One serious concern related to oral frailty is that elderly people often developed aspiration pneumonia”. You should better explain how oral frailty can develop aspiration pneumonia. What are the pathophysiological mechanisms? Can oral frailty cause aspiration pneumonia even in elderly people without neurodegenerative diseases? Is this pathological mechanism present in presbyphagia? The alteration of the oral phase of swallowing and the consequent premature spillage, can be the cause of increased risk of aspiration in the elderly people? You should better explain these mechanisms and add the related references.

Response 7: We appreciate your suggestion. We added an explanation of the relationship between oral frailty and aspiration pneumonia in the “Discussion” section.

Page 18, Lines 273-279

One serious concern related to oral frailty is that elderly people often developed aspiration pneumonia. Aspiration pneumonia is caused by many factors such as oral environment, swallowing dysfunction, decreased cough reflex, and immunodepression. These factors are associated with oral frailty [19]. Additionally, decreased tongue strength is associated with sarcopenia and contributes to oral frailty [20]. A reduction in tongue strength has been associated with an increased risk for aspiration as it increases the likelihood of bolus retention in the pharynx [21].

---

## [Decision Letter · Decision Letter 1]

6 Jul 2020

PONE-D-20-05211R1

Tongue thickness measured by ultrasonography is associated with tongue pressure in the Japanese elderly

PLOS ONE

Dear Dr. Nakamori,

Thank you for submitting your manuscript to PLOS ONE. After careful consideration, we feel that it has merit but does not fully meet PLOS ONE’s publication criteria as it currently stands. Therefore, we invite you to submit a revised version of the manuscript that addresses the points raised during the review process.

We look forward to receiving your revised manuscript.

Kind regards,

Giovanni Cammaroto

Academic Editor

PLOS ONE

Reviewers' comments:

Reviewer's Responses to Questions

**Comments to the Author**

1. If the authors have adequately addressed your comments raised in a previous round of review and you feel that this manuscript is now acceptable for publication, you may indicate that here to bypass the “Comments to the Author” section, enter your conflict of interest statement in the “Confidential to Editor” section, and submit your "Accept" recommendation.

Reviewer #2: All comments have been addressed

Reviewer #3: All comments have been addressed

2. Is the manuscript technically sound, and do the data support the conclusions?

Reviewer #2: Yes

Reviewer #3: Yes

3. Has the statistical analysis been performed appropriately and rigorously? 

Reviewer #2: Yes

Reviewer #3: Yes

4. Have the authors made all data underlying the findings in their manuscript fully available?

Reviewer #2: Yes

Reviewer #3: Yes

5. Is the manuscript presented in an intelligible fashion and written in standard English?

Reviewer #2: Yes

Reviewer #3: Yes

6. Review Comments to the Author

Reviewer #2: (No Response)

Reviewer #3: This is a very interesting manuscript with a good scientific method to develop. Statistics are ok. All data are provided properly, and language is correct.All data underlying the findings described in their manuscript are fully available without restriction.Language is correct clear and unambiguous.

I would like to introduce some issues to take into account with authors:

1)line 91 introduction what kind of paralysis?

Patients with tongue tie could not perform this test properly so they should be excluded.

otorhinolaryngologic disease (pharyngeal or laryngeal

tumor) (what happens with maxylofacial tumor if there is any trismus present you can not do the test. It will be beter history of head and neck neoplams.

2) Patients with sleep apnea syndrome had a different behavior with your conclusions. They have the lowest strength in the tongue and are correlated with increase tongue thickness. I will recommend you to read and include some reference to.

Wang SH, Keenan BT, Wiemken A, et al. Effect of Weight Loss on Upper Airway Anatomy and the Apnea-Hypopnea Index. The Importance of Tongue Fat. Am J Respir Crit Care Med. 2020;201(6):718-727 doi:10.1164/rccm.201903-0692OC

O'Connor-Reina C, Plaza G, Garcia-Iriarte MT, et al. Tongue peak pressure: a tool to aid in the identification of obstruction sites in patients with obstructive sleep apnea/hypopnea syndrome. Sleep Breath. 2020;24(1):281-286 doi:10.1007/s11325-019-01952-x

This kind of patients should be excluded from your study, or you should do some special emphasis in the discussion. OSAHS is another lifestyle related diseases commonly affect older patients. Your sample has normal BMI if they were overweight, do you think your conclusions will be fullfilled?.

3) There are many publications with IOPI to measure tongue strength in the world literature I suggested you could include some of them Their conclusions are similar as yours in the relationship with tongue strenght and age.

7. PLOS authors have the option to publish the peer review history of their article (what does this mean?). If published, this will include your full peer review and any attached files.

Reviewer #2: No

Reviewer #3: No

---

## [Author Response · Author response to Decision Letter 1]

10 Jul 2020

We sincerely appreciate the reviewer’s advice. Based on such suggestions, the manuscript has been revised as follows.

Reviewer #3:

This is a very interesting manuscript with a good scientific method to develop. Statistics are ok. All data are provided properly, and language is correct. All data underlying the findings described in their manuscript are fully available without restriction. Language is correct clear and unambiguous. I would like to introduce some issues to take into account with authors.

Response to the reviewer

Comment 1: line 91 introduction what kind of paralysis? Patients with tongue tie could not perform this test properly so they should be excluded. otorhinolaryngologic disease (pharyngeal or laryngeal

tumor) (what happens with maxylofacial tumor if there is any trismus present you can not do the test. It will be beter history of head and neck neoplams.

Response 1: We appreciate this comment. By ‘paralysis,’ we meant extremity paralysis, which is indicated in the manuscript (e.g. Line 92). Upon your suggestion, we excluded these patients from the study. Furthermore, we described the exclusion criteria in detail as follows: 

Page 6, Lines 89-93: We excluded patients who had a history of otorhinolaryngologic disease (brain, facial, neck, pharyngeal, or laryngeal tumors) and/or neurodegenerative disease. Patients with oral abnormalities, such as tongue-tie; maxillofacial dysfunction, such as trismus; and those with extremity paralysis were also excluded. 

Comment 2: Patients with sleep apnea syndrome had a different behavior with your conclusions. They have the lowest strength in the tongue and are correlated with increase tongue thickness. I will recommend you to read and include some reference to. Wang SH, Keenan BT, Wiemken A, et al. Effect of Weight Loss on Upper Airway Anatomy and the Apnea-Hypopnea Index. The Importance of Tongue Fat. Am J Respir Crit Care Med. 2020;201(6):718-727 doi:10.1164/rccm.201903-0692OCO' Connor-Reina C, Plaza G, Garcia-Iriarte MT, et al. Tongue peak pressure: a tool to aid in the identification of obstruction sites in patients with obstructive sleep apnea/hypopnea syndrome. Sleep Breath. 2020;24(1):281-286 doi:10.1007/s11325-019-01952-x. This kind of patients should be excluded from your study, or you should do some special emphasis in the discussion. OSAHS is another lifestyle related diseases commonly affect older patients. Your sample has normal BMI if they were overweight, do you think your conclusions will be fullfilled?.

Response 2: We thank the reviewer for these suggestions. There were 3 subjects in our study whose BMI was over 30 kg/m2 and who were not diagnosed with OSAHS. However, we did not perform a polysomnography on all subjects. Thus, this was indicated as a limitation of the study in the Discussion section. Upon your suggestion, we referred the articles recommended in our manuscript. We also considered racial difference, which was mentioned in the Discussion section.

Page 20, Lines 326-332: Third, we could not evaluate sleep apnea in the subjects of this study. It has been reported that people with sleep apnea have decreased tongue strength and increased tongue thickness due to excessive fat deposition, which causes tongue collapse [24,25]. In the present study, there were 3 subjects whose BMI was over 30 kg/m2 and were not diagnosed with obstructive sleep apnea hypopnea syndrome (OSAHS). However, we did not perform a polysomnography on all subjects. In future research, sleep apnea should be considered when studying the swallowing function.

Comment 3: There are many publications with IOPI to measure tongue strength in the world literature. I suggested you could include some of them. Their conclusions are similar as yours in the relationship with tongue strenght and age.

Response 3: We appreciate this suggestion. We mentioned the IOPI in the manuscript as follows:

Page 18, Lines 277-282: There are three main types of balloon-based tongue pressure measurement devices: the KayPENTAX device, the Iowa Oral Performance Instrument (IOPI), and the JMS Co. device. The JMS device, which is commonly used in Japan, shows lower values than the other balloon-based devices, but said values can be correlated linearly [6]. The IOPI is often used internationally, and shows the relationship between tongue pressure and age, which is consistent with our findings [19].

---

## [Editor Report · Decision Letter 2]

14 Jul 2020

Tongue thickness measured by ultrasonography is associated with tongue pressure in the Japanese elderly

PONE-D-20-05211R2

Dear Dr. Nakamori,

We’re pleased to inform you that your manuscript has been judged scientifically suitable for publication and will be formally accepted for publication once it meets all outstanding technical requirements.

Kind regards,

Giovanni Cammaroto

Academic Editor

PLOS ONE
---

## [Editor Report · Acceptance letter]

16 Jul 2020

PONE-D-20-05211R2 

Tongue thickness measured by ultrasonography is associated with tongue pressure in the Japanese elderly 

Dear Dr. Nakamori:

I'm pleased to inform you that your manuscript has been deemed suitable for publication in PLOS ONE. Congratulations! Your manuscript is now with our production department. 

Kind regards, 

on behalf of

Dr. Giovanni Cammaroto 

Academic Editor

PLOS ONE